# PianoMime: Learning a Generalist, Dexterous Piano Player from Internet Demonstrations

**Cheng Qian**
TU Munich

**Julen Urain**
TU Darmstadt

**Kevin Zakka**
UC Berkeley

**Jan Peters**
TU Darmstadt

**Abstract:** In this paper, we present PianoMime, a framework for training a piano-playing agent using Internet demonstrations. The Internet is a promising source of large-scale demonstrations for training our robot agents. In particular, in the case of piano playing, YouTube is full of videos of professional pianists playing a wide variety of songs. In our work, we leverage these demonstrations to train a generalist piano-playing agent capable of playing any song. Our framework is divided into three parts: a data preparation phase to extract the informative features from the YouTube videos, a policy learning phase to train song-specific expert policies from the demonstrations, and a policy distillation phase to distill the policies into a single generalist agent. We explore different policy designs for representing the agent and evaluate the influence of the amount of training data on the agent's ability to generalize to novel songs not present in the dataset. We show that we are able to learn a policy with up to 57% F1 score on unseen songs. Project website: https://pianomime.github.io/

**Keywords:** Imitation Learning, Reinforcement Learning, Dexterous Manipulation, Learning from Observations

## 1 Introduction

The Internet is a promising source of large-scale data for training generalist robot agents. If properly exploited, it is full of demonstrations (video, text, audio) of humans solving an infinite number of tasks [1, 2, 3] that could inform our robot agents on how to behave. However, learning from these databases is challenging for several reasons. First, unlike teleoperation demonstrations, video data does not specify the actions that the robot is performing, which typically requires the use of reinforcement learning to induce the robot's actions [4, 2, 5]. Second, videos typically show a human performing the task while the learned policy is applied to a robot. This often requires retargeting the human motion to the robot body [5, 6, 7]. Finally, as pointed out in [2], if we want to learn a generalist agent, we need to choose a task for which large databases are available and which allows for an unlimited variety of open-ended goals.

From opening doors [6] to manipulating ropes [8] or pick and place tasks [9, 10], previous work has successfully taught robot manipulation skills through observations. However, these approaches have been limited to robots with low dexterity or to a small variety of goals.

In this work, we focus on the task of **learning a generalist piano player from Internet demonstrations**. Piano-playing is a highly dexterous open-ended task [11]. Given two multi-fingered robot hands and a desired song, the goal of a piano-playing agent is to press the right keys, and only the right keys, at the right time. In addition, the task can be conditioned on arbitrary songs, allowing for large and high-dimensional goal conditioning.

In addition, the Internet is full of videos of professional piano players performing a wide variety of songs. Interestingly, these pianists often record themselves from above, making it easy to observe their performances. In addition, they usually share the MIDI files of the song they are playing, making it easier to extract relevant information.

8th Conference on Robot Learning (CoRL 2024), Munich, Germany.

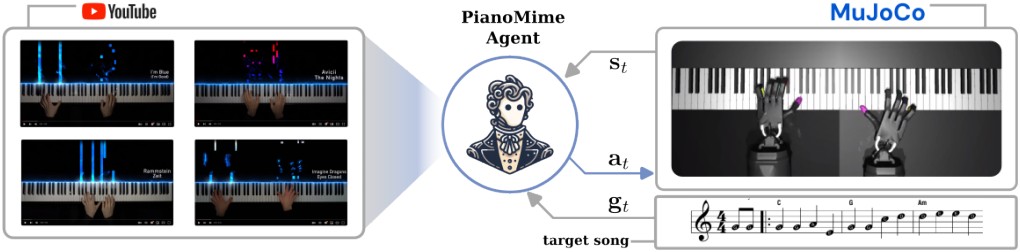

Figure 1: The goal of this work is to train a generalist piano-playing agent (PianoMime) from Youtube videos. We collect a set of videos and accompanying MIDI files and train a single agent to play any song, combining reinforcement learning and behavioral cloning.

To learn a generalist piano-playing agent from Internet data, we introduce **PianoMime**, a framework for training a single policy capable of playing any song (see Figure 1). In essence, the PianoMime agent is a goal-conditioned policy that generates actions in the configuration space, given the desired song to be played. At each time step, the agent receives a trajectory of keys to press as goal input. The policy then generates a trajectory of actions and executes them in chunks.

**To train the agent,** we combine both reinforcement learning and imitation learning. We train individual song-specific expert policies using reinforcement learning in conjunction with YouTube demonstrations, and we distill all the expert policies into a single generalist behavior cloning policy.

**To represent the agent,** we perform ablations of different architectural design strategies to model the behavior cloning policy. We investigate the benefit of incorporating representation learning to enhance the geometric information of the goal input. In addition, we explore the effectiveness of a hierarchical policy that combines a high-level policy generating fingertip trajectories with a learned *inverse model* generating joint space actions (see Figure 2). We show that the learned agent is able to play arbitrary songs not included in the training dataset with about 56% F1 score.

In summary, the main contribution of this work is a framework for training a generalist piano-playing agent using Internet demonstration data. To achieve this goal, we

- Introduce a method for learning policies from Internet demonstrations by decoupling the human movement information from the task-related information.

- Present a reinforcement learning approach that combines residual policy learning strategies [12, 13] with style reward-based strategies [5].

- Explore different policy architecture designs, introduce novel strategies to learn geometrically consistent latent features, and perform ablations on different architecture designs.

Finally, we release the dataset and trained models as a benchmark for testing Internet-data-driven dexterous manipulation.

## 2    Related Work

**Robotic Piano Playing.** Several studies have investigated the development of robots capable of playing the piano. In [14], multi-target Inverse Kinematics (IK) and offline trajectory planning are used to position the fingers over the intended keys. In [15], a Reinforcement Learning (RL) agent is trained to control a single Allegro hand to play the piano using tactile sensor feedback. However, the piano pieces used in these studies are relatively simple. Subsequently, in [11], an RL agent is trained to control two Shadow hands to play complex piano pieces by designing a reward function that includes a fingering reward, a task reward, and an energy reward. In contrast to previous approaches, our approach exploits YouTube piano-playing videos, allowing for faster training and more accurate robot behavior.

**Motion Retargeting and Reinforcement Learning.** Our work has similarities with motion retargeting [16], especially those works that combine motion retargeting with RL to learn control

policies [17, 18, 5, 19, 6]. Given a mocap demonstration, it is common to use the demonstration either as a reward function [5, 19] or as a nominal behavior for residual policy learning [18, 6]. In our work, we extract not only the mocap information, but also task-related information (piano states), which allows the agent to balance between mimicking the demonstrations and solving the task.

## 3   Method

The PianoMime framework consists of three phases: data preparation, policy learning, and policy distillation.

In the **data preparation phase**, given the raw video demonstration, we extract the informative signals needed to train the policies. Specifically, we extract the fingertip trajectories and a MIDI file that informs us of the state of the piano at each instant.

In the **policy learning phase**, we train song-specific policies via RL. This step is essential to generate the robot actions that are missing in the demonstrations. The policy is trained with two reward functions: a style reward and a task reward. The style reward aims to match the robot's finger movements with those of the human in the demonstrations to preserve the human style, while the task reward encourages the robot to press the right keys at the right time.

In the **policy distillation phase**, we train a single behavioral cloning policy to mimic all the song-specific policies. The goal of this phase is to train a single generalist policy that can play any song. We explore different policy designs and goal representation learning to improve the generalizability of the policy.

### 3.1   Data Preparation: From raw data to human and piano state trajectories

We generate the training dataset by web scraping. We download YouTube videos of professional piano artists playing different songs. In particular, we select YouTube channels that also upload MIDI files of the songs played. The MIDI files represent the trajectories of the piano's state (keys pressed/unpressed) throughout the song. We use the video to extract the movement of human pianists and the MIDI file to inform about the target state of the piano during the execution of the song. We choose the fingertip position as the key signal for the robot hand to mimic. While some dexterous tasks may require the use of the palm (e.g., grasping a bottle), we believe that mimicking the fingertip motion is sufficient for the piano-playing task. This also reduces the constraints on the robot, allowing it to adapt its embodiment more freely.

To extract the fingertip motion from the videos, we use MediaPipe [20], an open-source framework for perception. Given a frame from the demonstration videos, MediaPipe outputs the skeleton of the hand. We find that the classic top-view recording in YouTube videos of piano playing is highly beneficial for obtaining an accurate estimate of fingertip positions.

Note that since the videos are RGB, we lack the depth signal. Therefore, we predict the 3D fingertip positions based on the piano state. The detailed procedure is explained in Appendix A.

### 3.2   Policy Learning: Generating robot actions from observations

In the data preparation phase, we extract two trajectories: a human fingertip trajectory $\tau_x$ and a piano state trajectory $\tau_\flat$. The human fingertip trajectory $\tau_x : (x_1, \ldots, x_T)$ is a $T$-step trajectory of the 3D fingertip positions of two hands $x \in \mathbb{R}^{3 \times 10}$ (10 fingers). The piano state trajectory $\tau_\flat : (\flat_1, \ldots, \flat_T)$ is a $T$-step trajectory of piano states $\flat \in \mathbb{B}^{88}$, represented by an 88-dimensional binary variable representing which keys should be pressed.

Given the ROBOPIANIST [11] environment, **our goal** is to learn a goal-conditioned policy $\pi_\theta$ that plays the song defined by $\tau_\flat$ while matching the fingertip movement given by $\tau_x$. Note that satisfying both objectives jointly may be impossible. Perfectly tracking the fingertip trajectory $\tau_x$ might not lead to playing the song correctly. Although both trajectories are collected from the same source, errors in hand tracking and embodiment mismatches might lead to deviations, resulting in poor song performance. Therefore, we suggest using $\tau_x$ as a style guide behavior.

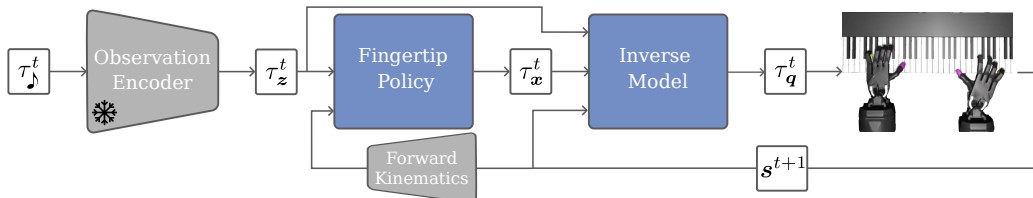

Figure 2: Proposed distillation policy architecture. Given a L steps window of a target song $\tau_\flat^t$ : ($\flat_{t:t+L}$) at time $t$, a latent representation $\tau_z^t$ is computed given a pre-trained observation encoder. Then, the policy is decoupled between a high-level fingertip policy that generates a trajectory of fingertip positions $\tau_x^t$ and a low-level inverse model that generates a trajectory of target joint position $\tau_q^t$.

Similar to [11], we formulate the piano playing as an **Markov Decision Process (MDP)** with the horizon of the episode $H$, which is the duration of the song to be played. The state observation is defined by the robot's proprioception $s$ and the goal state $g_t$. The goal state $g_t$ at time $t$ informs the desired piano key configurations $\flat$ in the future $g_t = (\flat_{t+1}, \ldots, \flat_{t+L})$, where $L$ is the lookahead horizon. As claimed in [11], to successfully learn how to play, the agent must be aware of several steps into the future to plan its actions. The action $a$ is defined as the desired configuration for both hands $q \in \mathbb{R}^{23 \times 2 + 1}$, each with 23 joint angles and one dimension for the sustain pedal.

We propose to solve the reinforcement learning problem by combining residual policy learning [12, 13, 6] and style mimicking rewards [5, 19].

**Residual Policy Architecture.** Given the fingertip trajectory $\tau_x$, we solve an IK [21] problem to obtain a trajectory of desired joint angles $\tau_q^{ik}$ : $(q_0^{ik}, \ldots, q_T^{ik})$ for the robot hands. Then we represent the policy $\pi_\theta(a|s, g_t) = \pi_\theta^r(a|s, g_t) + q_{t+1}^{ik}$ as a combination of a nominal behavior (given by the IK solution) and a residual policy $\pi_\theta^r$. Given the target state at time $t$, the nominal behavior is defined as the next desired joint angle $q_{t+1}^{ik}$. We then learn only the residual term around the nominal behavior. In practice, we initialize the robot at $q_0^{ik}$ and roll both the goal state and the nominal behavior with a sliding window along $\tau_\flat$ and $\tau_q^{ik}$ respectively.

**Style Mimicking Reward.** We also include a style-mimicking reward to preserve the human style in the trained robot actions. The reward function $r = r_\flat + r_x$ consists of a task reward $r_\flat$ and a style-mimicking reward $r_x$. While the task reward $r_\flat$ encourages the agent to press the correct keys, the style reward $r_x$ encourages the agent to move his fingertips similar to the demonstration $\tau_x$. We provide further details in Appendix D.

### 3.3 Policy Distillation: Learning a generalist piano-playing agent

In the policy learning phase, we train song-specific expert policies from which we roll out state and action trajectories $\tau_s$ : $(s_0, \ldots, s_T)$ and $\tau_q$ : $(q_0, \ldots, q_T)$. Then we generate a dataset $\mathcal{D}$ : $(\tau_s^i, \tau_q^i, \tau_x^i, \tau_\flat^i)_{i=1}^N$ where $N$ is the number of songs learned. Given the dataset $\mathcal{D}$, we apply Behavioral Cloning (BC) to learn a single generalist piano-playing agent $\pi_\theta(q_{t:t+L}, x_{t:t+L}|s_t, \flat_{t:t+L})$, which outputs configuration space actions $q_{t:t+L}$ and fingertip movements $x_{t:t+L}$ conditioned on the current state $s_t$ and the future desired piano states $\flat_{t:t+L}$.

We explore different strategies to represent and learn the behavioral cloning policy and improve its generalization capabilities. In particular, we explore (**1**) representation learning approaches to induce spatially informative features, (**2**) a hierarchical policy structure for sample-efficient training, and (**3**) expressive generative models [22, 23, 24] to capture the multimodality of the data. Also, inspired by current behavioral cloning approaches [22, 25], we train policies that output sequences of actions rather than single-step actions and execute them in chunks.

**Representation Learning.** We pre-train an observation encoder over the piano state $\flat$ to learn spatially consistent latent features. We hypothesize that two piano states that are spatially close should lead to latent features that are close. Using these latent features as a target should lead to better

generalization. To obtain the observation encoder, we train an autoencoder with a reconstruction loss over a Signed Distance Field (SDF) defined on the piano state. Specifically, the encoder compresses the binary vector of the goal into a latent space, while the decoder predicts the SDF function value of a randomly sampled query point (the distance between the query point and the closest "on" piano key). For the BC policy, we concatenate L-timestep desired piano states and pass through the pre-trained observation encoder to obtain the latent goal representation. We provide more details in Appendix G.

**Hierarchical Policy.** We represent the piano-playing agent with a hierarchical policy. The high-level fingertip policy takes a sequence of desired future piano states $\flat$ and outputs a trajectory of human fingertip positions $x$. Then, a low-level inverse model takes the fingertip and piano state trajectories as input and outputs a trajectory of desired joint angles $q$. On the one hand, while fingertip trajectory data is readily available from the Internet, obtaining low-level joint trajectories requires solving a computationally expensive RL problem. On the other hand, while the high-level mapping ($\flat \mapsto x$) is complex and involves fingerings, the low-level mapping ($x \mapsto q$) is relatively simple, involving a task space to configuration space mapping. This decoupling allows us to train the more complex high-level mapping on large, cheap datasets, and the simpler low-level mapping on smaller, expensive datasets. We visualize the policy in Figure 2.

**Expressive Generative Models.** Considering that the human demonstration data of piano playing is highly multimodal, we explore the use of expressive generative models to better represent this multimodality. We compare the performance of different deep generative models based policies, such as Diffusion Policies [22] and Behavioral Transformer [23], as well as a deterministic policy.

## 4 Experimental Results

We divide the experimental evaluation into three parts. In the first part, we investigate the performance of our proposed framework in learning song-specific policies via RL. In the second part, we perform ablation studies on policy designs for learning a generalist piano-playing agent by distilling the previously learned policies via BC. Finally, in the third part, we study the influence of the amount of training data on the generalization capabilities.

**Dataset and Evaluation Metrics** All experiments are performed on our collected dataset, which contains the notes and corresponding demonstration videos and fingertip trajectories of 60 piano songs from a Youtube channel, **PianoX** [1]. To standardize the length of each task, each song is divided into several clips, each 30 seconds long (the dataset contains a total of 431 clips, 258K state-action pairs). In addition, we select 12 unseen clips to investigate the generalization ability of the generalist policy. These clips consist of completely new songs that do not appear in the training dataset. We use the same evaluation metrics from RoboPianist [11], i.e., precision, recall, and F1 score (see Appendix B). We run each policy for the whole song and evaluate its performance.

**Simulation Environment** Our experimental setup uses the ROBOPIANIST simulation environment [11], implemented in the Mujoco [26]. The agent predicts target joint angles at 20Hz, and the targets are converted to torques using PD controllers running at 500Hz. The notes of the songs are also discretized at a frequency of 20Hz. We use the same setup as [11] with two modifications: 1) The z-axis sliding joints attached to both forearms are enabled to allow more versatile hand movements. 2) We increase the proportional gain of the PD controller for the x-axis sliding joints to allow faster horizontal movement, which we feel is essential for some fast-paced piano songs.

### 4.1 Evaluation on learning song-specific policies from demonstrations

In this section, we evaluate the song-specific policy learning and aim to answer the following questions: (**1**) Does the integration of human demonstrations with RL help to achieve better performance? (**2**) Which elements of the learning algorithm are most important for good performance?

We use Proximal Policy Optimization (PPO) [27] because we found that it performs best compared to other RL algorithms. We compare our model to two baselines:

---

[1]https://www.youtube.com/channel/UCsR6ZEA0AbBhrF-NCeET6vQ

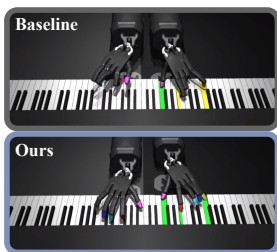
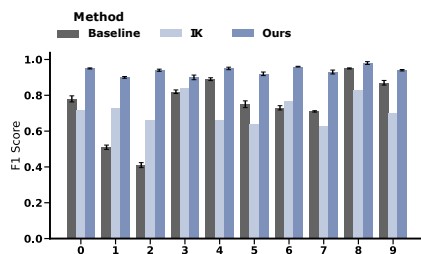
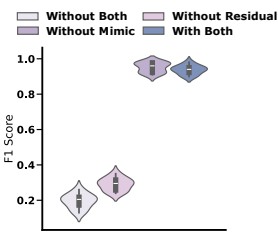

Figure 3: Left: Hand postures (Baseline and Ours). Middle: The F1 score achieved by three methods for 10 chosen clips; Right: The F1 score achieved by excluding different elements in RL.

**RoboPianist [11]** We use the RL method introduced in [11]. We keep the same reward functions as in the original work and manually label the fingering from the demonstration videos to provide the fingering reward.

**Inverse Kinematics (IK) [21]** Given a fingertip trajectory demonstration $\tau_x$, a Quadratic Programming-based IK solver [21] is used to compute a target, joint position trajectory and execute it open-loop.

We select 10 clips from the collected dataset with different levels of difficulty. We individually train specialized policies for each of the 10 clips using both the baselines and our method. We then evaluate and compare their performance based on the obtained F1 score.

**Performance.** As shown in Figure 3, our method consistently outperforms the RoboPianist baseline for all 10 clips, achieving an average F1 score of 0.94 compared to the baseline's 0.74. We attribute this improvement to the incorporation of human priors, which narrows the RL search space to a favorable subspace, thereby encouraging the algorithm to converge on more optimal policies. In addition, the IK method achieves an average F1 score of 0.70, only

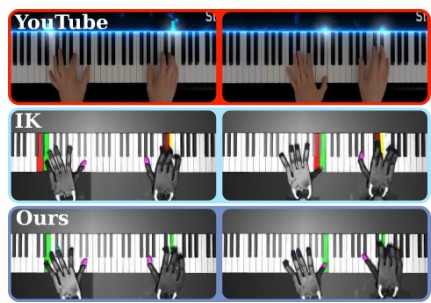

Figure 4: Comparison of hand poses. Top: Youtube video, Middle: IK solution given the video. Bottom: After residual RL.

slightly lower than the baseline. This demonstrates the effectiveness of incorporating human priors, which provides a strong starting point for RL. We also observe that our method trains faster than the baseline. On an RTX 4090, the baseline took an average of 4 hours to train, while our method took an average of 2.5 hours.

**Impact of Elements.** Our RL method has two main elements: a style-mimicking reward and a residual learning. We exclude each element individually to study their respective influences on policy performance (see Figure 3). We clearly observe the critical role of residual learning, which implies the benefit of using human demonstrations as nominal behavior. We observe a marginal performance increase of 0.03 when excluding the style-mimicking reward, but this also results in a larger discrepancy between the robot and human fingertip trajectories. Thus, the weight of the style-mimicking reward can be considered as a parameter that controls the human similarity of the learned robot actions. The ablation study for this weight is discussed in Appendix F.

**Hand Pose Visualization.**[2] We provide Figure 3 and Figure 4 as an example of hand poses in different settings and provide attached videos on the website with further examples. In Figure 4, we exemplify that our policy places the hands in similar poses to the YouTube videos. We measure the distance between fingertips in YouTube videos and the robot in Appendix F. We observe that the IK nominal behavior leads the robot to place the fingers in positions similar to those in YouTube videos. The RL policy then slightly adapts the fingertip positions to press the keys correctly. Besides, we observe that the RoboPianist baseline sometimes presents visually unhuman-like motions. For ex-

---

[2]The key colors in Figure 3 and Figure 4 mean the following: green indicates a correctly pressed key, yellow indicates a key that should be pressed but is not, and red indicates a key that should not be pressed but is.

| | | Multi-RL | BC-MSE | AIRL | **Two-Stage Diff** | -res | w/o SDF | One-Stage | BeT |
|---|---|---|---|---|---|---|---|---|---|
| **Train** | P | 0.85 | 0.56 | 0.83 | 0.87 | **0.89** | 0.86 | 0.53 | 0.63 |
| | R | 0.20 | 0.29 | 0.24 | 0.78 | **0.80** | 0.76 | 0.34 | 0.42 |
| | F1 | 0.12 | 0.30 | 0.21 | 0.81 | **0.82** | 0.78 | 0.35 | 0.49 |
| **Test** | P | **0.95** | 0.54 | 0.91 | 0.69 | 0.71 | 0.66 | 0.58 | 0.53 |
| | R | 0.18 | 0.22 | 0.20 | 0.54 | **0.55** | 0.49 | 0.27 | 0.30 |
| | F1 | 0.13 | 0.21 | 0.17 | 0.56 | **0.57** | 0.51 | 0.26 | 0.31 |

Table 1: Quantitative results evaluated on Training and Test Datasets. Test datasets consist of 12 clips unseen in the training dataset. We report Precision (P), Recall (R) and F1-score (F1).

ample, in Figure 3 Left, the middle finger and the ring finger of the left robot hand are at relatively unhuman-like positions.

## 4.2 Evaluation of model design strategies for policy distillation

This section focuses on the evaluation of the policy distillation for playing different songs. We evaluate the influence of different policy design strategies on the agent's performance. We aim to assess (**1**) the impact of integrating a pre-trained observation encoder to induce spatially consistent features, (**2**) the impact of a hierarchical design of the policy, and (**3**) the performance of different generative models on piano-playing data.

**Proposed Models.** We propose two base policies, `Two-stage Diff` and `Two-stage Diff-res` policy. Both use hierarchical policies and goal representation learning, as described in Section 3.3. The only difference between them is that the low-level policy of `Two-stage Diff` predicts the target joints directly, while `Two-stage Diff-res` predicts the residual term of an IK solver. A detailed description of the policies can be found in Appendix I.

**Baselines.** We consider as baselines a `Multi-task` RL policy and a BC policy with MSE Loss from [11]. Additionally, we implement an Adversarial Inverse Reinforcement Learning (AIRL) baseline [28]. We provide further details of the models in Appendix I.

**Ablation Models.** To analyze the impact of our policy design choices, we design three variants of our proposed model, i.e., `w/o SDF`: We train a policy that directly receives the 88-dimensional binary representation of the goal, without using the SDF observation encoder, to evaluate the impact of the goal's representation learning, `One-stage`: We train an end-to-end diffusion policy to evaluate the impact of the hierarchical architecture, `BeT`: We train a two-stage Behavior-Transformer [23] to evaluate the impact of using diffusion models.

**Results.** As shown in Table 1, our methods (`Two-stage Diff` and `Two-stage Diff-res`) outperform the others on both training and test datasets. `Multi-task` RL and AIRL have higher precision on the test dataset, but this is because they barely press any keys. We observe a large improvement when using both diffusion policies instead of `BeT`, a hierarchical policy instead of an end-to-end policy and a slight improvement when using a pre-trained observation encoder, especially on the test dataset.
We also observe a slight performance improvement when the model predicts the residual term of IK (`Two-stage Diff-res`).

## 4.3 Evaluations on the impact of the data in the generalization

In this section, we investigate the impact of scaling the training data on the generalization capabilities of the agent. We divide the experiments into two parts:
(**1**) We evaluate the impact of scaling the training data on the performance of three policy designs (`One-stage Diff`, `Two-stage Diff`, and `Two-stage Diff-res`) evaluated on the test dataset by training them with different proportion of the dataset (see Figure 5 Top).

**(2)** We evaluate the influence of a good high-level policy of `Two-stage Diff` by training different high-level policies on different proportions of the dataset (see Figure 5 Bottom). We provide additional results in Appendix M.

**Impact of scaling training data.** We observe that both `Two-stage Diff` and `Two-stage Diff-res` show consistent performance improvement when increasing the training data (Figure 5 Top). This trend implies that the two-stage policies have not yet reached their performance saturation with the given data and could potentially continue to benefit from additional training data in future works.

**Impact of high-level policy quality.** We further employ different combinations of the high-level and low-level policies of `Two-stage Diff` trained with different proportions of the dataset and assess their performance. In addition, we introduce a high-level oracle policy that outputs the ground-truth fingertip positions from the human demonstration videos. The results (see Figure 5 Bottom) demonstrate that the overall performance of the policy is significantly influenced by the quality of the high-level policy. Low-level policies paired with Oracle high-level policies consistently outperform the ones paired with other high-level policies. Besides, we observe early performance convergence with increasing training data when paired with a low-quality high-level policy.

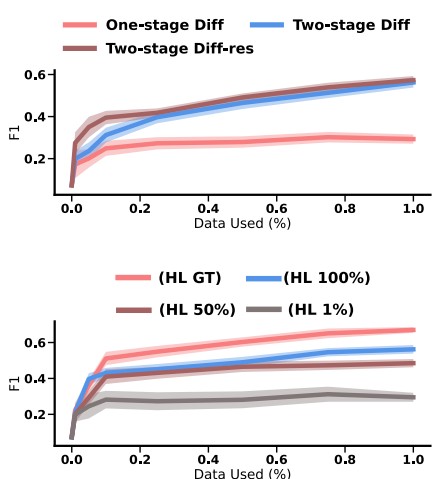

Figure 5: F1 scores in test data with varying amounts of training data. **Top**: Performance comparison for different policy designs. **Bottom**: Performance comparison on models trained with different proportions of high-level and low-level datasets. The x-axis represents the percentage of the low-level dataset utilized, while HL % indicates the percentage of the high-level dataset used.

## 4.4 Limitations

**Inference Speed** One of the limitations is the inference speed. The models operate with an inference frequency of approximately 15Hz on an RTX 4090 machine, which is lower than the standard real-time demand on hardware. Future works can employ faster diffusion models, e.g., DDIM [29], to speed up the inference.

**Out-of-distribution Data** Most of the songs in our collected dataset are of modern style. When evaluating the model on the dataset from [11], which mainly contains classical songs, the performance degrades. It implies the model's limited generalization across songs of different styles. Future work can collect more diverse training data to improve this aspect.

**Acoustic Experience** Although the policy achieves up to 57% F1-score on unseen songs, we found that higher accuracy is still necessary to make the song acoustically appealing and recognizable. Future work should focus on improving this accuracy to enhance the overall acoustic experience.

## 5 Conclusion

In this work, we present PianoMime, a framework for training a generalist robotic pianist using Internet video sources. The proposed framework is composed of three distinct phases: first, extract task-related and human motion-related trajectories from videos, second, train song-specific policies with reinforcement learning and finally, distill all the song-specific policies in a single generalist policy. We found that the resulting policy demonstrates an impressive generalization capability, achieving an average F1-score of 57% on unseen songs. We believe that the findings for learning fine motor skills in piano playing can be applied to other tasks that require high dexterity and precision, and scenarios where robot data collection through teleoperation is challenging.

**Acknowledgments**

We are extremely grateful to Jan Brüdigam and Stefan Sosnowski from the Chair of Information-oriented Control, Technical University of Munich for their advicing. We would also like to thank the YouTube channel PianoX for providing excellent demonstration videos on piano playing.

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

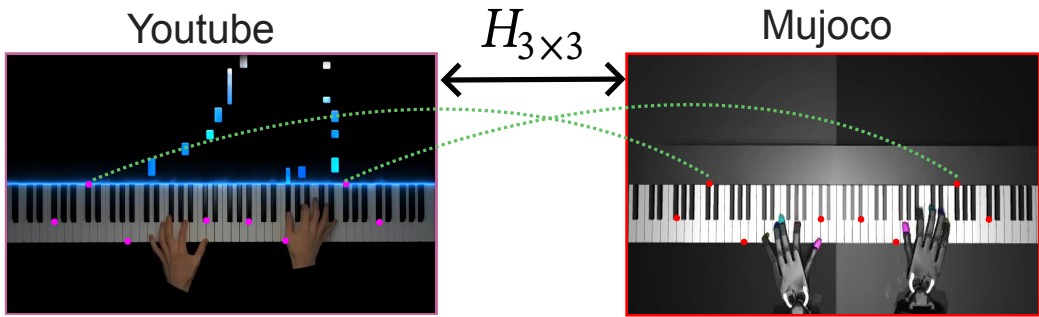

Figure 6: Compute homography matrix given 8 correspondence feature points.

## A    Retargeting: From human hand to robot hand

To retarget from the human hand to the robot hand, we follow a structured process.

**Step 1: Homography Matrix Computation** Given a top-view piano demonstration video, we firstly choose $n$ different feature points on the piano. These points could be center points of specific keys, edges, or other identifiable parts of the keys that are easily recognizable (see Figure 6). Due to the uniform design of the pianos, these points represent the same physical positions in both the video and Mujoco. Given the chosen points, we follow the Eight-point Algorithm to compute the Homography Matrix $H$ that transforms the pixel coordinate in videos to the x-y coordinate in Mujoco (the z-axis is the vertical axis).

**Step 2: Transformation of Fingertip Trajectory** We then obtain the human fingertip trajectory with MediaPipe [20]. We collect the fingertips positions every 0.05 seconds. Then we transform the human fingertip trajectory within pixel coordinate into the Mujoco x-y 2D coordinate using the computed homography matrix $H$.

**Step 3: Heuristic Adjustment for Physical Alignment** We found that the transformed fingertip trajectory might not physically align with the notes, which means there might be no detected fingertip that physically locates at the keys to be pressed or the detected fingertip might locate at the border of the key (normally a human presses the middle point on the horizontal axis of the key). This misalignment could be due to the inaccuracy of the hand-tracking algorithm and the homography matrix. Therefore, we perform a simple heuristic adjustment on the trajectory to improve the physical alignment. Specifically, at each timestep of the video, we check whether there is any fingertip that physically locates at the key to be pressed. If there is, we adjust its y-axis value to the middle point of the corresponding key. Otherwise, we search within a small range, specifically the neighboring two keys, to find the nearest fingertip. If no fingertip is found in the range or the fingertip has been assigned to another key to be pressed, we then leave it. Otherwise, we adjust its y-axis value to the center of the corresponding key to ensure proper physical alignment.

**Step 4: Z-axis Value Assignment** Lastly, we assign the z-axis value for the fingertips. For the fingertips that press keys, we set their z-axis values to $0$. For other fingertips, we set their z-axis value to $2 \cdot h_{key}$, where $h_{key}$ is the height of the keys in Mujoco.

## B    Evaluation Metrics

We use the same metrics from RoboPianist [11], i.e., Precision, Recall, and F1 score. Here we provide a detailed definitions of them:

- **True Positive (TP):** Keys that should be pressed are pressed.
- **False Positive (FP):** Keys that should not be pressed are pressed.

- **False Negative (FN):** Keys that should be pressed are not pressed.

$$\text{Precision} = \frac{TP}{TP + FP} \tag{1}$$

$$\text{Recall} = \frac{TP}{TP + FN} \tag{2}$$

$$\text{F1} = \frac{2 \cdot \text{Precision} \cdot \text{Recall}}{\text{Precision} + \text{Recall}} \tag{3}$$

Given the ground truth and the executed piano state trajectory, we calculate Precision, Recall, and F1 score for each timestep. We then get the overall Precision, Recall, and F1 score by averaging them over timesteps. In this way, precision evaluates the robot's capability of avoiding pressing the wrong keys, while recall evaluates the robot's capability of pressing the correct keys. F1 score combines both of them.

## C   Implementation of Inverse Kinematics Solver

The implementation of the IK solver is based on the approach of [21]. The solver addresses multiple tasks simultaneously by formulating an optimization problem and finding the optimal joint velocities that minimize the objective function. The optimization problem is given by:

$$\min_{\dot{q}} \sum_i w_i \| J_i \dot{q} - K_i v_i \|^2, \tag{4}$$

where $w_i$ is the weight of each task, $K_i$ is the proportional gain and $v_i$ is the velocity residual. We define a set of 10 tasks, each specifying the desired position of one of the robot's fingertips. We do not specify the desired quaternions. All the weights $w_i$ are set to be equal. We use quadprog [3] to solve the optimization problem with quadratic programming. The other parameters are listed in Table 2.

Table 2: The parameters of IK solver

| Parameter | Value |
|---|---|
| Gain | 1.0 |
| Limit Gain | 0.05 |
| Damping | 1e-6 |
| Levenberg-Marquardt Damping | 1e-6 |

## D   Detailed MDP Formulation of Song-specific Policy

We present a detailed representation of the reward functions applied in our method in Table 3.

## E   Training Details of Song-specific Policy

We use PPO [27] (implemented by StableBaseline 3 [30]) to train the song-specific policy with residual RL(See Algorithm 1). All of the experiments are conducted using the same network architecture and tested using 3 different seeds. Both actor and critic networks are of the same architecture, containing 2 MLP hidden layers with 1024 and 256 nodes, respectively, and GELU [31] as activation functions. The detailed hyperparameters of the networks are listed in Table 6.

---

[3]https://github.com/quadprog/quadprog

Table 3: The detailed reward function to train the song-specific policy. The Key Press reward is the same as in [11], where $k_s$ and $k_g$ represent the current and the goal states of the key respectively, and g is a function that transforms the distances to rewards in the [0, 1] range. $p_{df}$ and $p_{rf}$ represent the fingertip positions of human demonstrator and robot respectively.

| Reward | Formula | Weight | Explanation |
|--------|---------|--------|-------------|
| Key Press | $0.5 \cdot g(\|k_s - k_g\|_2) + 0.5 \cdot (1 - \mathbf{1}_{\text{false positive}})$ | 2/3 | Press the right keys and only the right keys |
| Mimic | $g(\|p_{df} - p_{rf}\|_2)$ | 1/3 | Mimic the demonstrator's fingertip trajectory |

Table 4: The observation space of song-specific agent.

| Observation | Unit | Size |
|-------------|------|------|
| Hand and Forearm Joint Positions | Rad | 52 |
| Hand and forearm Joint Velocities | Rad/s | 52 |
| Piano Key Joint Positions | Rad | 88 |
| Piano key Goal State | Discrete | 88 |
| Demonstrator Forearm and Fingertips Cartesian Positions | m | 36 |
| Prior control input $\tilde{u}$ (solved by IK) | Rad | 52 |
| Sustain Pedal state | Discrete | 1 |

## F    Ablation Study for Weight of Style-mimicking Reward

To numerically evaluate the human-likeness of the robot motion, we include an additional metric, $\Delta ft$, which computes the average Euclidean distance between the robot fingertip positions and the human demonstrators for each timestep. We further make an ablation study to explore the impact of the weight of style-mimicking reward on $\Delta ft$ and F1 score, respectively (See Figure 7). The result indicates that when the weight of the mimic reward is zero, the F1 score is the highest, but the

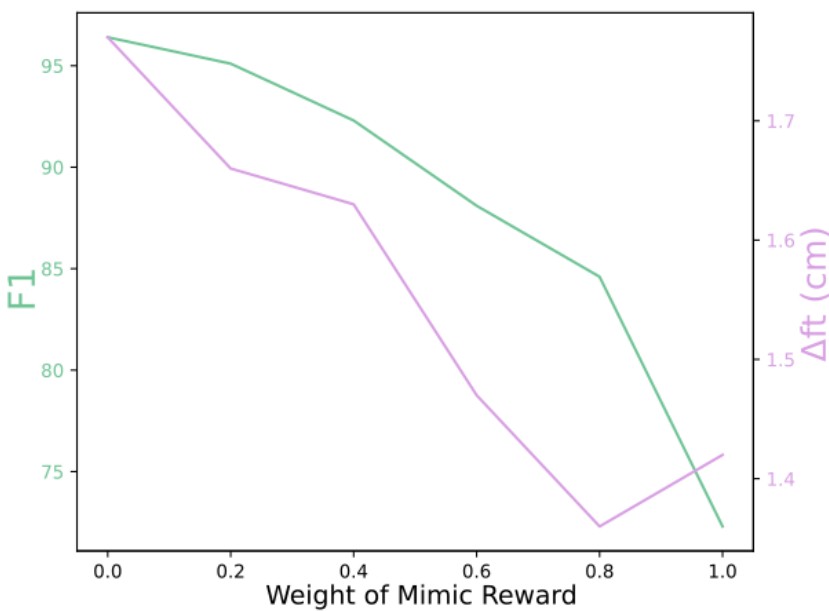

Figure 7: Impact of the weight of style-mimicking reward on $\Delta ft$ and F1 score

Table 5: The action space of song-specific agent.

| Action | Unit | Size |
|---|---|---|
| Target Joint Positions | Rad | 46 |
| Sustain Pedal | Discrete | 1 |

Table 6: The Hyperparameters of PPO

| Hyperparameter | Value |
|---|---|
| Initial Learning Rate | 3e-4 |
| Learning Rate Scheduler | Exponential Decay |
| Decay Rate | 0.999 |
| Actor Hidden Units | 1024, 256 |
| Actor Activation | GELU |
| Critic Hidden Units | 1024, 256 |
| Critic Activation | GELU |
| Discount Factor | 0.99 |
| Steps per Update | 8192 |
| GAE Lambda | 0.95 |
| Entropy Coefficient | 0.0 |
| Maximum Gradient Norm | 0.5 |
| Batch Size | 1024 |
| Number of Epochs per Iteration | 10 |
| Clip Range | 0.2 |
| Number of Iterations | 2000 |
| Optimizer | Adam |

relative distance between the human fingertip positions and the robot's fingertip positions is also the greatest. As we increase the influence of the mimic reward, the performance decreases, while the relative distance to the human fingertip positions also diminishes. This allows us to balance between improving performance and achieving a behavior more similar to the videos by adjusting the mimic reward. The discrepancy is unavoidable since the robot's embodiment differs from that of a human, and accurately playing the piano song might necessitate some deviation from human behavior.

## G Representation Learning of Goal

We train an autoencoder to learn a geometrically continuous representation of the goal (See Figure 8 and Algorithm 2). During the training phase, the encoder $\mathcal{E}$, encodes the original 88-dimensional binary representation of a goal piano state $\flat_t$ into a 16-dimensional latent code $z$. The positional encoding of a randomly sampled 3D query coordinate $x$ is then concatenated with the latent code $z$ and passed through the decoder $\mathcal{D}$. We use positional encoding here to represent the query coordinate more expressively. The decoder is trained to predict the SDF $f(x, \flat_t)$. We define the SDF value of $x$ with respect to $\flat_t$ as the Euclidean distance between the $x$ and the nearest key that is supposed to be pressed in $\flat_t$, mathematically expressed as:

$$\text{SDF}(x, \flat_t) = \min_{p \in \{p_i | \flat_{t,i} = 1\}} \|x - p\|, \tag{5}$$

where $p_i$ represents the position of the $i$-th key on the piano. The encoder and decoder are jointly optimized to minimize the reconstruction loss:

$$L(x, , \flat_t) = (\text{SDF}(x, \flat_t) - \mathcal{D}(\mathcal{E}(v, x)))^2. \tag{6}$$

**Algorithm 1:** Training of the song-specific policy with residual RL

---

1: Initialize actor network $\pi_\theta$
2: Initialize critic network $v_\phi$
3: **for** $i = 1 : N_{iteration}$ **do**
4:    # Collect trajectories
5:    **for** $t = 1 : T$ **do**
6:       Get human demonstrator fingertip position $x_t$ and observation $o_t$
7:       Compute the prior control signal that tracks $x_t$ with the IK controller $\tilde{u}_t = ik(x_t, o_t)$
8:       Run policy to get the residual term $r_t = \pi_\theta(o_t)$
9:       Compute the adapted control signal $u_t = \tilde{u}_t + r_t$
10:      Execute $u_t$ in environment and collect $s_t, u_t, r_t, s_{t+1}$
11:    **end for**
12:    # Update networks
13:    **for** $n = 1 : N$ **do**
14:       Sample a batch of transitions $\{(s_j, u_j, r_j, s_{j+1})\}$ from the collected trajectories
15:       Update the actor and critic network with PPO
16:    **end for**
17: **end for**

---

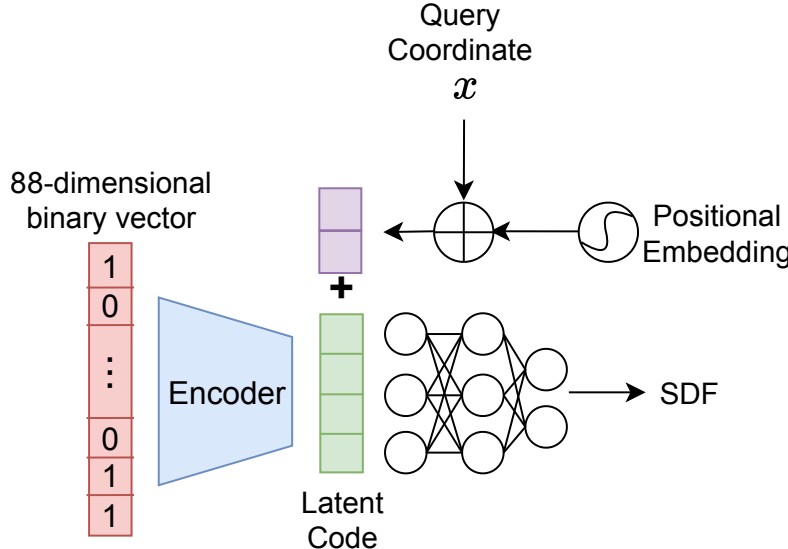

Figure 8: 1) Encoding: The encoder compresses the binary representation of the goal into latent code. 2) Decoding: A 3D query coordinate $x$ is randomly sampled. A neural network predicts the SDF value given the positional encoding of $x$ and the latent code.

We pre-train the autoencoder using the **GiantMIDI** dataset [4], which contains 10K piano MIDI files of 2,786 composers. The pre-trained encoder maps the $\flat_t$ into the 16-dimensional latent code, which serves as the latent goal for behavioral cloning. The encoder network is composed of four 1D-convolutional layers, followed by a linear layer. Each successive 1D-convolutional layer has an increasing number of filters, specifically 2, 4, 8, and 16 filters, respectively. All convolutional layers utilize a kernel size of 3. The linear layer transforms the flattened output from the convolutional layers into a 16-dimensional latent code. The decoder network is an MLP with 2 hidden layers, each with 16 neurons. We train the autoencoder for 100 epochs with a learning rate of $1e - 3$.

---

[4]https://github.com/bytedance/GiantMIDI-Piano

---

**Algorithm 2:** Training of the goal autoencoder

---
1: Initialize encoder $\mathcal{E}_\phi$
2: Initialize decoder $\mathcal{D}_\psi$
3: **for** $i = 1 : N_{epoch}$ **do**
4:     **for** $j = 1 : N_{batch}$ **do**
5:         **for** each goal $v$ in batch **do**
6:             Compute the latent code $\mathbf{z} = \mathcal{E}_\psi(\flat_t)$
7:             Sample a 3D coordinate as query $\mathbf{x} = $ Sample3DCoordinate()
8:             Compute the positional encoding of query $\mathbf{pe} = $ PositionalEncoding$(x)$
9:             Compute the output of the decoder conditioned by the query $\mathcal{D}_\phi(z, pe)$
10:            Compute the SDF value of query SDF$(x, \flat_t)$
11:            Compute the reconstruction loss $L$
12:        **end for**
13:        Compute the sum of the loss
14:        Compute the gradient
15:        Update network parameter $\phi, \psi$
16:    **end for**
17: **end for**

---

# H   Training Details of Diffusion Model

All the diffusion models utilized in this work, including One-stage Diff, the high-level and low-level policies of Two-stage Diff, Two-stage Diff-res, and Two-stage Diff w/o SDF, share the same network architecture. The network architecture is the same as the U-net diffusion policy in [22] and optimized with DDPM [32], except that we use temporal convolutional networks (TCNs) as the observation encoder, taking the concatenated goals (high-level policy) or fingertip positions (low-level policy) of several timesteps as input to extract the features on the temporal dimension. Each level of U-net is then conditioned by the outputs of TCNs through FiLM [33].

High-level policies take the goals over 10 timesteps and the current fingertip position as input and predict the human fingertip positions. In addition, we add a standard Gaussian noise on the current fingertip position during training to facilitate generalization. We further adjust the y-axis value of the fingertips pressing the keys in the predicted high-level trajectories to the midpoint of the keys. This adjustment ensures closer alignment with the data distribution of the training dataset. Low-level policies take the predicted fingertip positions, the goals over 4 timesteps, and the proprioception state as input to predict the robot's actions. The proprioception state includes the robot joint positions and velocities, as well as the piano joint positions. We use 100 diffusion steps during training. To achieve high-quality results during inference, we find that at least 80 diffusion steps are required for high-level policies and 50 steps for low-level policies.

Table 7: The Hyperparameters of DDPM

| Hyperparameter | Value |
|---|---|
| Initial Learning Rate | 1e-4 |
| Learning Rate Scheduler | Cosine |
| U-Net Filters Number | 256, 512, 1024 |
| U-Net Kernel Size | 5 |
| TCN Filters Number | 32, 64 |
| TCN Kernel Size | 3 |
| Diffusion Steps Number | 100 |
| Batch Size | 256 |
| Number of Iterations | 800 |
| Optimizer | AdamW |
| EMA Exponential Factor | 0.75 |
| EMA Inverse Multiplicative Factor | 1 |

## I  Policy Distillation Experiment

**Two-stage Diff.** The model consists of a hierarchical policy with a pre-trained goal observation encoder, as described in Section 3.3. Note that the entire dataset is used for training the high-level policy, while only around 40 % of the collected clips (110K state-action pairs) are trained with RL and further used for training the low-level policy. The detailed network implementation is described in Appendix H.

**Two-stage Diff w/o SDF.** We directly use the binary representation of the goal instead of the SDF embedding representation to condition the high-level and low-level policies.

**Two-stage Diff-res** The model is close to Two-stage Diff, with slight changes. We employ an IK solver to compute the target joints given the fingertip positions predicted by the high-level policy. The low-level policy predicts a residual term around the IK solution instead of the robot's actions.

**Two-stage BeT.** We train both high-level and low-level policies with Behavior Transformer [23] instead of DDPM. The hyperparameter of Bet are listed in Table 8.

**One-stage Diff.** We train a single diffusion model to predict the robot actions given the SDF embedding representation of goals and the proprioception state.

**Multi-task RL.** We create a multi-task environment where for each episode a random song is sampled from the dataset. The observation and action space, as well as the reward function of the environment, follow the same settings as described in [11]. Consequently, we use Soft-Actor-Critic (SAC) [34] to train a single agent within the environment. Both the actor and critic networks are MLPs, each with 3 hidden layers, and each hidden layer contains 256 neurons.

**BC-MSE.** We train a feedforward network to predict the robot action of the next timestep conditioned on the binary representation of goal and proprioception state with MSE loss. The feedforward network is an MLP with 3 hidden layers, each with 1024 neurons.

**AIRL [28].** We use the same multi-task environment as Multi-task RL. We use an open-source implementation of AIRL based on PPO [5], where the actor and critic networks in PPO are

---
[5]https://github.com/toshikwa/gail-airl-ppo.pytorch

MLPs, each consisting of three hidden layers with 256 neurons per layer. The reward and shaping term of the discriminator also use the same MLP architecture, with three hidden layers and 256 neurons in each layer. We collect expert state-action pairs by rolling out the song-specific policies, where the state consists of the song's notes and proprioceptive state, and the action is the joint-space robot action. The collected data is then fed into the discriminator.

Table 8: The Hyperparameters of Behavior Transformer

| Hyperparameter | Value |
|---|---|
| Initial Learning Rate | 3e-4 |
| Learning Rate Scheduler | Cosine |
| Number of Discretization Bins | 64 |
| Number of Transformer Heads | 8 |
| Number of Transformer Layers | 8 |
| Embedding Dimension | 120 |
| Batch Size | 256 |
| Number of Iterations | 1200 |
| Optimizer | AdamW |
| EMA Exponential Factor | 0.75 |
| EMA Inverse Multiplicative Factor | 1 |

## J   F1 Score of All Trained Song-Specific Policies

Figure 10 shows the F1 score of all song-specific policies we trained.

## K   Detailed Results on Test Dataset

In Table 9 and Table 10, we show the Precision, Recall, and F1 score of each song in our collected test dataset and the Etude-12 dataset from [11], achieved by Two-stage Diff and Two-stage Diff-res, respectively. We observe an obvious performance degradation when testing on Etude-12 dataset. We suspect that the reason is due to out-of-distribution data, as the songs in the Etude-12 dataset are all classical, whereas our training and test dataset primarily consists of modern songs.

Table 9: Quantitative results of each song in our collected test dataset

| Song Name | Two-stage Diff | | | Two-stage Diff-res | | |
|---|---|---|---|---|---|---|
| | Precision | Recall | F1 | Precision | Recall | F1 |
| Forester | 0.81 | 0.70 | 0.68 | 0.79 | 0.71 | 0.67 |
| Wednesday | 0.66 | 0.57 | 0.58 | 0.67 | 0.54 | 0.55 |
| Alone | 0.80 | 0.62 | 0.66 | 0.83 | 0.65 | 0.67 |
| Somewhere Only We Know | 0.63 | 0.53 | 0.58 | 0.67 | 0.57 | 0.59 |
| Eyes Closed | 0.60 | 0.52 | 0.53 | 0.61 | 0.45 | 0.50 |
| Pedro | 0.70 | 0.58 | 0.60 | 0.67 | 0.56 | 0.47 |
| Ohne Dich | 0.73 | 0.55 | 0.58 | 0.75 | 0.56 | 0.62 |
| Paradise | 0.66 | 0.42 | 0.43 | 0.68 | 0.45 | 0.47 |
| Hope | 0.74 | 0.55 | 0.57 | 0.76 | 0.58 | 0.62 |
| No Time To Die | 0.77 | 0.53 | 0.55 | 0.79 | 0.57 | 0.60 |
| The Spectre | 0.64 | 0.52 | 0.54 | 0.67 | 0.50 | 0.52 |
| Numb | 0.55 | 0.44 | 0.45 | 0.57 | 0.47 | 0.48 |
| **Mean** | **0.69** | **0.54** | **0.56** | **0.71** | **0.55** | **0.57** |

Table 10: Quantitative results of each song in the Etude-12 dataset

| Song Name | Two-stage Diff | | | Two-stage Diff-res | | |
|---|---|---|---|---|---|---|
| | Precision | Recall | F1 | Precision | Recall | F1 |
| FrenchSuiteNo1Allemande | 0.45 | 0.31 | 0.34 | 0.39 | 0.27 | 0.30 |
| FrenchSuiteNo5Sarabande | 0.29 | 0.23 | 0.24 | 0.24 | 0.18 | 0.19 |
| PianoSonataD8451StMov | 0.58 | 0.52 | 0.52 | 0.60 | 0.50 | 0.51 |
| PartitaNo26 | 0.35 | 0.22 | 0.24 | 0.40 | 0.24 | 0.26 |
| WaltzOp64No1 | 0.44 | 0.31 | 0.33 | 0.43 | 0.28 | 0.31 |
| BagatelleOp3No4 | 0.45 | 0.30 | 0.33 | 0.45 | 0.28 | 0.32 |
| KreislerianaOp16No8 | 0.43 | 0.34 | 0.36 | 0.49 | 0.34 | 0.36 |
| FrenchSuiteNo5Gavotte | 0.34 | 0.29 | 0.33 | 0.41 | 0.31 | 0.33 |
| PianoSonataNo232NdMov | 0.35 | 0.24 | 0.25 | 0.29 | 0.19 | 0.21 |
| GolliwoggsCakewalk | 0.60 | 0.43 | 0.45 | 0.57 | 0.40 | 0.42 |
| PianoSonataNo21StMov | 0.32 | 0.22 | 0.25 | 0.36 | 0.23 | 0.25 |
| PianoSonataK279InCMajor1StMov | 0.43 | 0.35 | 0.35 | 0.53 | 0.38 | 0.39 |
| **Mean** | **0.42** | **0.31** | **0.33** | **0.43** | **0.30** | **0.32** |

## L  Failure Cases

For song-specific policies, because the starting position of the hands is fixed to the middle of the piano, we observe that some policies do not behave well at the beginning of the song. Particularly, when they are required to press the keys on the sides of the piano. For multi-song policies, especially for unseen songs, we observe that while the robot tends to press the desired keys, it sometimes wrongly presses the neighboring ones. This likely occurs because the model does not accurately learn the system dynamics.

# M   Extension: Evaluations on the impact of the data in the generalization

In this section, we provide additional details on Section 4.3. We present the recall, precision, and F1 scores for the two experiments conducted in Section 4.3 in Figure 9. By observing the recall and precision, we can clearly observe that increasing the dataset positively impacts both the precision and recall of the learned policy. This indicates that the robot not only presses the proper keys more often (improves recall) but also avoids pressing the wrong keys equally often (improves precision). Thus, the observed improvement of the F1-score is led by both.

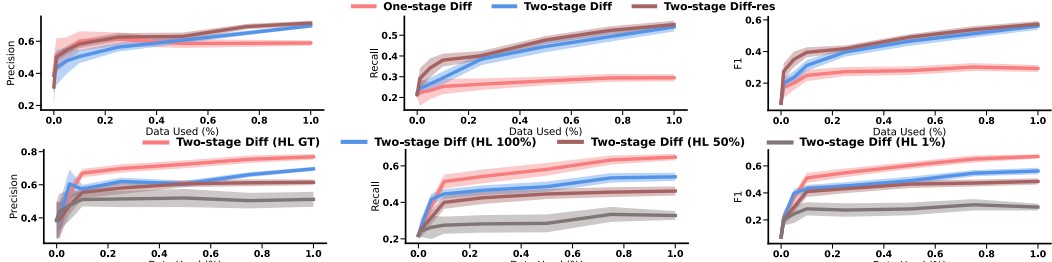

Figure 9: Precision, Recall, and F1 Score for policies trained with varying amounts of data volumes evaluated on the test dataset. **Top**: The models (One-Stage diffusion, Two-Stage Diffusion, and Two-Stage Diffusion-res) are trained with the same proportion of high-level and low-level datasets. **Bottom**: Two-stage diffusion models are trained with different proportions of high-level and low-level datasets. The x-axis represents the percentage of the low-level dataset utilized, while HL % indicates the percentage of the high-level dataset used.

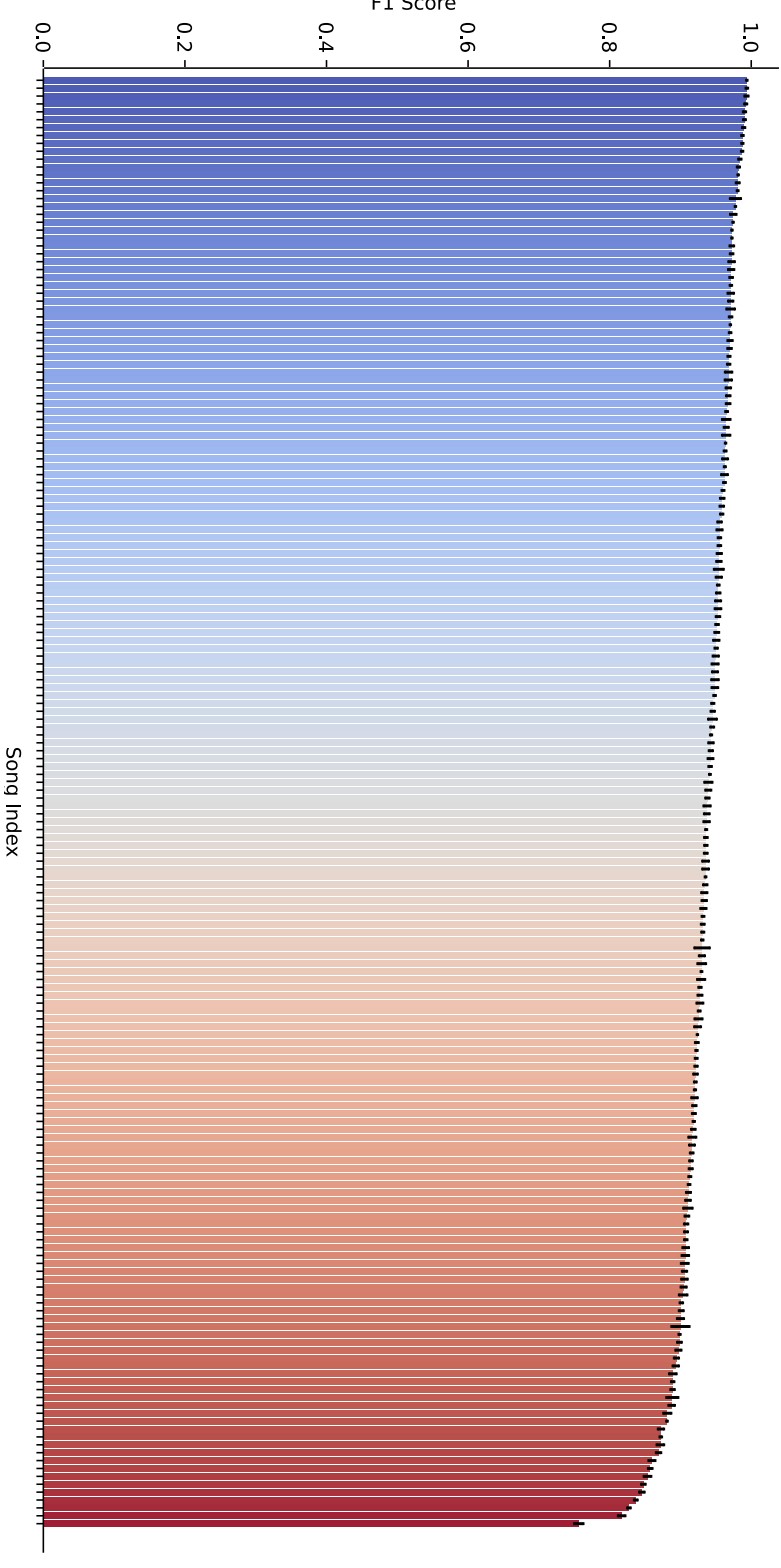

Figure 10: F1 score of all 184 trained song-specific policies (descending order)

