# OpenReview forum: "PianoMime: Learning a Generalist, Dexterous Piano Player from Internet Demonstrations"
_robot-learning.org/CoRL/2024/Conference — CoRL 2024_

### Official Review · Reviewer_gJnu · 2024-07-20

**Originality:** 2
**Technical Quality:** 2
**Clarity Of Presentation:** 2
**Potential Impact:** 2
**Recommendation:** 3
**Confidence:** 3

**Review:**

Strengths:

-The results are generally positive, and ablations are included, such as removing distillation, stages, and SDF. However, it is unclear whether there is still an encoder.

-The paper is easy to read, though, there are missing details noted below.

-The design of the algorithmic system is intuitive and simple.

Weaknesses:

-Please see the questions below, as the paper is omitting important details to help review the paper.

-The goal representation learning is primarily a neural network encoder using a Signed Distance Field (SDF). Removing the SDF makes the approach a little bit worse, but it is unclear what "w/o SDF" means precisely.

-Overall, the paper appears to be a combination of known, existing techniques and does not have an innovation at the algorithmic level but rather at a systems level. Ablation studies support the design of the system, so there is an incremental, positive improvement.

-It would have been helpful to evaluate the approach in the real-world.

-The approach is missing a baseline for inverse reinforcement learning from the raw data; a baseline for DAgger during policy distillation would also help as would using RL fine-tuning after distillation.

-The qualitative results (Figure 3) are not insightful other than to show that "ours" is better than a baseline. More insights into failure cases would be useful.

-The metrics used in this paper appear to be one-step prediction metrics for precision, recall, and F1-score. It is unclear precisely how these metrics are computed. An issue here, though, might be that the metrics assume "teacher forcing" in that the robot may be put into the correct state at time t, then the model asked to predict and implement the next sound (unclear what exactly is being measured), and then that is it. It would be much more informative to actually have the robot play an entire MIDI file open-loop as if were a real performance and then measure the distance between the ground-truth MIDI file and then the MIDI file that the robot produced. I imagine the performance degradation would be significant and would give a much better assessment of which approach is actually best and playing a song.

-The paper does not appear to measure how well the robot captures the finger movement / style of the human player and instead seems to focus on notes; however, this is a bit unclear. For example, a human-expert eval to compare the style of the robot's play for anthropomorphic properties would be helpful.

Writing:

-Some formatting issues exist, such as no spacing between paragraphs in Section 3.1.

================ Post-Rebuttal =============================

Based upon the rebuttal, I have decided to move to a weak accept.

**Quality Of The Limitations Section:**

2

**Questions For Rebuttal:**

-The specifics of the music trajectory are unclear. There are orders of magnitude difference in the wall-clock time between a whole note tied to another whole note in a next bar vs. a 64th note. The MIDI file should account for things like staccato and slur, but the discretization of time here is key. It would be helpful if more details are provided.

-Why was imitation learning (DAgger) not used instead of BC?

-The performance improvement by doing distillation is quite modest and not uniformly better. Why is that, and why then include it as a contribution?

-What would happen if you further fine-tuned the distilled policy using RL?

-It would have been helpful to report precisely how precision, recall, and F1-score are being reported. Also, are these averaged over all classes? In the case of class imbalance, this may skew the results and make it harder to understand when/where the model performs well or not.

-In Figure 2, is the observation encoder only receiving one discrete note unit at time, t, or is it receiving a sequence (i.e., the goal)?

**Robotics Focus:**

3

**Summary Of Paper:**

This paper proposes a framework, PianoMime, that can learn from demonstration and reinforcement learning, to play on piano a variety of songs in a simulated environment. The policy is hierarchical in that it takes as input a sequence of notes to play (MIDI File) and outputs fingertip positions for the robot; next, the fingertip positions are passed to a network that outputs the joint commands, q. The paper compares to baselines in a standard environment and shows strongly positive results against baselines.

**Summary Of Recommendation:**

The paper is lacking numerous details; the work is positive but more of a collection of existing techniques; and it may not have employed sufficiently robust evaluation metrics.

---

### Official Review · Reviewer_CxL8 · 2024-07-29
**This work has novel learning system design for teaching dexterous robotic hands to play piano.**

**Originality:** 4
**Technical Quality:** 3
**Clarity Of Presentation:** 2
**Potential Impact:** 3
**Recommendation:** 3
**Confidence:** 3

**Review:**

This work provides a novel framework for piano playing with the combination of song-specific RL policies and generalized BC policies. Learning from human demonstrations on YouTube with top-down views seems natural and efficient to leverage human priors. I believe that the hierarchical system structure can be potentially extended to other dexterous manipulation learning.

The experiments are thorough and the authors properly evaluated all modules in the system. However, in the Appendix, the authors said there were errors in F1 scores for all methods in the paper. I am concerned about whether the statements from the results will still hold with updated scores.

The paper clarification needs a lot of improvement. I would suggest the authors to proofread the draft. Some issues are listed here:
- Please be consistent about terminologies in the paper such as "Internet" vs "internet".
- It seems “Inverse Kinematics”, “Reinforcement Learning”, “Markov Decision Process”, “Behavioral Cloning” are highlighted and linked to empty links.
- What does “cross domain inverse dynamics model” mean? Is it "inverse kinematics model"? I don’t think there is anything about dynamics here since only kinematics are used to map ee poses to the joint poses.
- What is “FK” in Figure 2? Is it “Forward Kinematics”?
- Broken sentences:
“At each timestep, the agent receives as goal input a trajectory of the keys to be pressed.”
“we combine both reinforcement learning with imitation learning ”
“Given a mocap demonstration, it has been common to exploit the demonstration rather as a reward function [5, 19] or as a nominal behavior for residual policy learning [18, 6]. ”
“We select the fingertip position as the essential signal to mimic with the robot hand ”

**Quality Of The Limitations Section:**

2

**Questions For Rebuttal:**

- What are the major failure modes for the policy distillation? I understand that the test data (unseen clips) might be out-of-distribution, but how different are they from the training data?
- What do the colors, i.e. green, yellow, and red represent in the visualization?
- Please also report the training time and potentially compare it with baseline methods to support the statement: "In contrast with previous approaches, our approach exploits YouTube piano-playing videos, enabling faster training and more accurate and human-like robot behavior.".
- Please correct the F1 scores and double-check the analysis of the experiments.

**Robotics Focus:**

3

**Summary Of Paper:**

This work uses human demonstrations from internet to teach two dexterous robotic hands to play piano in simulation. The authors introduced a RL-based song-specific policy learning, and further presented a generalizable policy learning with behavior cloning. The proposed learning framework has been tested on unseen songs and can achieve 57%  F1 score.

**Summary Of Recommendation:**

I recommend to accept the paper for the novel system design and thorough experiments. The authors addressed my questions and concerns during the rebuttal.

---

### Official Review · Reviewer_Hkur · 2024-07-31
**This paper presents a novel framework for teaching agents to play the piano using YouTube video demonstrations. It shows significant improvements in performance and human-likeness through innovative methods like residual policy architecture and style-mimic rewards. While the paper is well-structured, the analysis is complex and somewhat difficult to follow. Overall, the work makes a substantial and original contribution to robot learning.**

**Originality:** 4
**Technical Quality:** 5
**Clarity Of Presentation:** 3
**Potential Impact:** 3
**Recommendation:** 3
**Confidence:** 3

**Review:**

Quality: The paper is well-structured, presenting a comprehensive framework for learning piano playing through YouTube video demonstrations. The methodology is robust, and the results are thoroughly analyzed, despite the complexity.

Clarity: The paper is generally clear and well-organized. However, the analysis section is somewhat overwhelming due to the multitude of factors considered, making it difficult to trace the results easily.

Originality: The work is highly original, leveraging publicly available online demonstrations to teach agents complex tasks like playing the piano. The use of hierarchical policies and goal representation learning to improve generalization is particularly innovative.

Significance: This work is significant as it advances the field of robot learning by demonstrating a practical application of reinforcement learning and online demonstrations. It has the potential to impact future research and practical applications in robotics and artificial intelligence.

Strengths:
Innovative Approach: Utilizing YouTube videos for training agents is a novel and creative idea.
Improved Performance: The residual policy architecture and style-mimic reward clearly enhance performance and human-likeness.
Generalization: The method effectively generalizes policies using hierarchical policies and goal representation learning.
Clear Structure: The paper is well-organized, with a consistent narrative and clear presentation of the methodology and results.

Weaknesses:
Complex Analysis: The analysis of the results is dense and somewhat difficult to follow due to the numerous factors involved.
Overwhelming Data: The extensive data and variables considered in the analysis can be overwhelming for the reader, potentially obscuring key findings.
Reasoning: Why/how better agents learning how to play piano could be beneficial to other robot learining areas could be discussed more.

Overall, this paper presents a significant and original contribution to the field of robot learning, with clear improvements in performance and generalization. However, simplifying the analysis or providing clearer explanations could enhance the clarity and accessibility of the results.

**Quality Of The Limitations Section:**

3

**Questions For Rebuttal:**

The performance of the model during reinforcement learning (RL) shows that the version without the mimic component achieves a better F1 score. Later, the paper discusses the style-mimicking reward as a parameter influencing the human-like quality of the robot's actions. To strengthen the argument, the paper should include a metric or evaluation demonstrating the significance of the style-mimicking reward and its optimal weight. Specifically, it would be beneficial to quantify how much the mimic score impacts the overall performance and determine the appropriate weight for this reward.

Additionally, the paper would benefit from incorporating evaluations involving real people to assess the performance of the piano-playing agent. This would provide a practical metric for the effectiveness of the learned robot actions and validate the human-likeness of the performance in a real-world context.

**Robotics Focus:**

3

**Summary Of Paper:**

The paper utilized online piano playing demonstrations to extract relevant data, developed song-specific policies through reinforcement learning (RL), and subsequently employed behavior cloning techniques to enhance generalization across different pieces (including unseen songs).

**Summary Of Recommendation:**

Based on the evaluation, I recommend accepting the paper with minor revisions. The paper presents an innovative framework for teaching agents piano playing using YouTube videos, with notable improvements in performance and human-likeness through advanced techniques like residual policy architecture and style-mimic rewards. However, the analysis is complex and challenging to follow, and the paper lacks a user study involving real humans. Additionally, it does not clearly demonstrate how learning to play the piano can benefit other areas of robot learning. Addressing these issues will enhance the paper's clarity and broader relevance.

---

### Author Rebuttal · Authors · 2024-08-06

In the following we list all the changes we did in the manuscript:
- We corrected grammar and vocabulary errors.
- We adapted section 4.2 and section 4.3 to enhance readability and make the experimental analysis clearer.
- We added additional missing key clarifications (training time, colors of the keys, song discretization frequency …).
- We updated our F1 scores to fix the error we highlighted in the Appendix and added details on how they are computed.
- We added an ablation study in Appendix F exploring the influence of the style-mimicking reward weight
- We added an Adversarial Inverse Reinforcement Learning baseline using PPO.
- We used DeepL and Grammarly and proofread the paper, to correct grammar errors.

---

### Decision · Program_Chairs · 2024-09-04

**Decision:**

Accept

**Comment:**

**Paper summary**

This paper presents a method for learning piano-playing policies from internet videos as demonstrations. The results indicate that the proposed method is generalizable to new songs and performs well in comparison to baseline RL methods.

**Review summary**

Summary of strengths:
+ The paper addresses an important challenge: leveraging the large amounts of data available through internet videos for robot policy learning.
+ The evaluation demonstrates that the proposed method is effective in modeling human-likeness, performance, and generalization from the training data.
+ The evaluation is thorough and is performed over multiple ablations of the system.

Summary of weaknesses:
- The reviewers (CxL8 and gJnu) note several key clarifications that are necessary for fully understanding the paper. The evaluation analysis needs more explanation and structure (Reviewer Hkur). **[This has been addressed by the rebuttal and revised paper.]**
- The results may not be accurate due to an error in computing F1 scores, as noted in the appendix. The method for computing these metrics is also missing. **[This has been addressed by the rebuttal and revised paper.]**
- The evaluation would be improved with a real-world evaluation and an additional baseline (inverse RL from raw data). **[The authors have added AIRL as a baseline.]**


**Response to rebuttal**

Two of the reviewers have increased their recommendations from weak reject to weak accept, stating that their concerns were addressed by the author rebuttal. The third reviewer maintains their recommendation of weak accept.

**Recommendations for improvement**

* The paper would be further strengthened by the inclusion of DAgger as a baseline.
* The human-subjects study should not be included in this paper. The authors are encouraged to revisit this in future work with IRB approval and a more rigorous survey scale and analysis.